# Case Studies for Dangerous Dust Explosions in South Korea during Recent Years

**Seonggyu Pak [1,†], Seongho Jung [2,†], Changhyun Roh [3,4,5] and Chankyu Kang [6,*]**

[1]  Korea Occupational Safety and Health Agency, Major Accident Prevention Center in Chungju, 85 Daerimro, Chungju, Chungbuk 27477, Korea

[2]  Department of Environmental and Safety Engineering, Ajou University, Worldcupro 206, Yeongtong-gu, Suwon 16499, Korea

[3]  Decommissioning Technology Research Division, Korea Atomic Energy Research Institute (KAERI), 989-111 Daedukdaero, Yuseong, Daejeon 34057, Korea

[4]  Advanced Radiation Technology Institute (ARTI), Korea Atomic Energy Research Institute (KAERI), Jeonbuk 56212, Korea

[5]  Quantum Energy Chemical Engineering, University of Science and Technology, 217, Gajeong-ro, Yuseong-gu, Daejeon 34113, Korea

[6]  Ministry of Labor and Employment, Chemical Accident Prevention Division, 422 Hannuridae-ro, Sejong 30117, Korea

*  Correspondence: chemnet75@korea.kr; Tel.: +82-44-200-2556

†  These authors contributed equally to this work.

**Abstract:** Despite recent extensive research and technical development to prevent and mitigate dust explosions, processes that produce and handle combustible materials in the form of powders and dusts, either as a main product or as an undesired by-product, have become a constant dust explosion threat as they become more sophisticated and complicated. This study analyzed the characteristics of 53 dust explosions that occurred in South Korea over the last 30 years, and investigated the differences of dust explosions that happened in various countries, such as Japan, the United States, the United Kingdom, and France. In addition, case studies showed the severity of dust explosions occurring in South Korea. Through the special focus on the three most recent years of dust explosions, the causes and processes of the accidents were identified. Analyses of dust explosions in South Korea show that they were mainly caused by organic matter and metal, and, unfortunately, dust explosions occurred repeatedly during grinding, mixing, and injection of powder materials into facilities. No reported accidents occurred during the production processes of wood or paper during the last three years. Taking these characteristics into account, effective ways to prevent or mitigate dust explosions at workplaces where many dust explosions occurred were suggested.

**Keywords:** dust explosions; facility; prevent; mitigate; organic matter; metal

## 1. Introduction

Many industrial dust explosions occur in powder-processing equipment during crushing, grinding, conveying, storing, and other processes [1]. These operations are still widely used in industrial plants and will be applied to new fields of industry, potentially triggering many dust explosions in the future [2]. As a result, the threat of dust explosions remains a continued risk [3]. Many kinds of industrial plants use powder manufacturing facilities, which produce several types of dust known to be explosive, and cause catastrophic loss of life and property [4]. In a plant environment where finely ground dust spreads in the air, dust explosions may occur immediately, depending on the source of ignition if certain conditions (i.e., minimum explosive concentration (MEC)) are reached. The

types of dust that cause dust explosions are wood, paper products, grain and food, metal and metal products, power generation, coal mining, and textile manufacturing. Factors such as dust properties, dust particle size, and the formation of combustion by-products vary the speed and extent of flame propagation and have a complex mechanism in that they include simultaneous momentum, energy, and mass transport in reactive multi-phase systems [5,6]. In addition, it is a phenomenon involving a rapid increase in pressure and heat, accompanied by sound, which is caused by the fact that the substance burns at a very high rate [7].

Dust explosions seriously destroy human beings, properties, and the environment. Among the classification of dusts, relative dust explosion severity is classified by the $K_{st}$ index, divided into four groups from non-explosive (i.e., St 0) to highly explosive (i.e., St 3) [8]. Several factors, such as dust concentration, the composition of the dust and moisture content, particle size and shape, type of dust, and turbulence in the system, affect the degree of ignition and propagation of flame in the dust cloud [9,10]. This means that substances with a high risk of dust explosion (i.e., smaller sizes and thin flats of particles, metallic particles, a dust cloud with less oxygen content) need special attention during process control. Recently, several dust explosions have caused serious damage: (i) 14 deaths and 38 injuries occurred in the Imperial Sugar refinery in the United States in 2008 [11]; (ii) an accident occurred at a maize starch plant in China in 2010, killing 21 employees and injuring 47 [12]; (iii) an HDPE (high density polyethylene) dust explosion occurred in the silo at the Yeosu Industrial Complex in South Korea, resulting in the deaths of six workers who were in the process of installing a manhole, and injuring an additional 11 workers on 14 March, 2013; (iv) an aluminum dust deflagration occurred at an auto parts factory in Jiangsu, China, on 2 August, 2014, resulting in a catastrophic 146 deaths and 114 injured [13], with the investigation concluding that the bag filters in the dust collection bags had to be cleaned by mechanical shaking at regular intervals, but the primary deflagration had begun with the shaking system, which was out of operation for a long period of time; (v) in 2015, 10 people were killed and 485 were injured by an explosion of a colored, starch-based dust at a party called "Color Play Asia" in Taiwan [14]. Among various types of dusts, there have been many studies on aluminum dust because it is the most explosive among metal dusts, and accounted for about 25% of the metal dusts from 1980 to 2005 in the United States [15–17]. The temperature of metal dust flame is more than 1000 K higher than organic dust flame, which causes an increase in pressure. The risk of dust explosion as a result of increased temperatures and pressure has contributed to metal dust receiving more attention [18]. Meanwhile, coal dust explosions were the most frequent and serious accidents in China, with 106 cases reported between 1949 and 2007 [19]. Fabiano et al. (2014) analyzed 98 incident statistics related to coal between 2000 and 2011 using data bank FACTS, managed by the Unified Industrial and Harbor Fire Department of Rotterdam–Rozenburg, NL [20]. Coal and coal dust caused 171 and 39 fatalities, respectively, and classifying the major accident scenario corresponded to explosions (56.0%), followed by fires (30.8%) and releases (13.2%). Analysis of the incidents revealed that two-thirds of them occurred in power and process plants due to human error and technical failure/malfunction of a component or equipment.

To create a dust explosion, the necessary conditions for dust explosions are well represented by the explosive pentagon: (i) Fuel, (ii) oxidant, (iii) ignition source, (iv) mixing of the fuel and oxidant, and (v) confinement of the resulting mixture [21]. There must be a certain density of dispersed dust in the air so that the dust accumulates. A confined space maintains sufficient pressure for the primary dust explosion to occur [22]. The consequence of the secondary dust explosion causes disastrous and uncontrolled explosions, which occur when the blast wave from the primary explosion entrains the dust layer and propagates through the plant. The dust becomes the fuel of the emerging flame, leading to extensive explosions due to large amounts of dust and the high energy of ignition [23].

Several controversial issues have recently been solved to provide efficient preventive and mitigatory measures. Farrell et al. (2013) conducted an experiment with a large-scale test rig, consisting of a 5 m$^3$ vented vessel connected to two 400 mm diameter pipelines, to study dust explosion propagation. This study showed that the dust explosion was able to move against a process flow at a long distance [24].

Vogl and Radandt (2005) carried out the experiment and concluded that powder handling plants should be cautious with design and operation because of the possibility of dust explosion propagation, even with small-diameter pipes [25]. Valiulis et al. (1999) and Farrell et al. (2013) concluded that a dust explosion can propagate if there is little to no dust at all in the connected pipes [24,26]. Recently, the development of advanced numerical models has played an important role in providing guidelines on the stable and optimized design of process equipment to mitigate or prevent dust explosions [27]. However, controversial subjects, such as explosion venting and secondary explosions, are still unsatisfactory because these issues are subject to increased complexity and process variability due to process automation.

The purpose of this study is to provide information on dust explosions that occurred in South Korea, and to compare them with dust explosions in Japan, the United States, the United Kingdom, and France. The characteristics of dust explosions are presented through analysis of their causes over 30 years in South Korea. Case studies of catastrophic dust explosions in South Korea present the severity and damage of the accidents. Special focus on the last three years of dust explosions is to identify the causes and process of dust explosions, and to suggest effective ways to prevent or mitigate them in workplaces. To the best of our knowledge, this is the first study to introduce several cases of dust accidents and suggest preventive measures because there is little awareness and little information about dust explosions in South Korea.

## 2. Dust Explosions in South Korea from 1984 to 2018

### 2.1. Dust Explosions for South Korea

From 1984 to 2018, 53 combustible dust- and flammable solid-related fire explosions were reported in South Korea. Figure 1 identifies dust explosions by type of industry and their main cause. According to the analysis of dust explosions based on the types of industries shown in Figure 1a, 30% of all dust explosions occurred in the metal industry, 25% in the chemical industry, and 21% in the food industry. Dust explosions in wood and paper industries were as low as 9%. As reflected in the industrial structure, the major cause of dust explosions, shown in Figure 1b, was metal, which accounted for 44% of all dust explosions. Plastic and food sources each cause 19% of dust explosions. The number of accidents due to combustible metal dusts was the highest, but explosions caused by plastics accounted for about 50% of deaths and about 47% of injuries. Deaths due to weakly explosive plastic dusts were over three times more than those of strong explosive metal dusts, presumably due to the establishment of government safety guidelines and industrial characteristics.

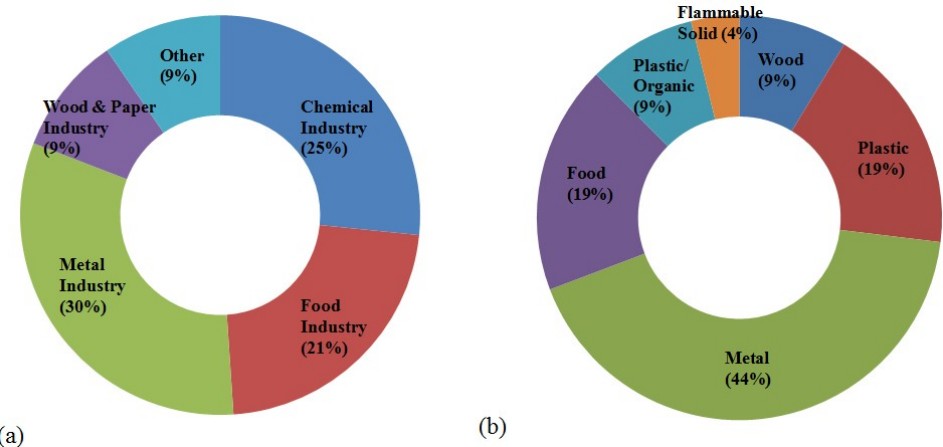

**Figure 1.** Analysis of dust explosions in South Korea. (**a**) Occurrence of dust explosions by type of industry; (**b**) Cause of dust explosions.

## 2.2. Recent Dust Explosions

Table 1 presents examples of dust explosions from May 2015 to May 2018. Dust explosions caused by organic matters and metal caused significant property loss as well as human loss. Analysis of major accident processes reveals that accidents occurred repeatedly during grinding, mixing, and injection of powder materials into facilities, while no accidents occurred in the production process of wood or paper during the last three years.

**Table 1.** Dust explosion in South Korea (May 2015 to May 2018).

| Year/Month | Equipment | Hazard Material | Accident Process |
|---|---|---|---|
| 2018/5 | Rocket propulsion plant | Ammonium perchlorate | Grinding |
| 2018/5 | Dust collector | Zirconium oxide | Grinding |
| 2018/4 | Dissolution | Tungsten oxide alloy | Injection of metal powder |
| 2018/4 | Alloy Manufacturing | Co–Cr alloy | Manufacture of alloy powder |
| 2017/7 | Silo | Polypropylene | PP pellet transfer |
| 2016/1 | Mixer | Propanoic acid | Mixing process |
| 2015/10 | Silo | Terephthalic acid | Welding and cutting of manhole |
| 2015/9 | Chemical Manufacturing | Aluminum oxide mixture | Mixing process |
| 2015/3 | Reactor | Terephthalic acid | Injection of Terephthalic acid into reactor |
| 2015/5 | Melting furnace | Waste Aluminum | Injection of waste aluminum into furnace |

## 3. Case Studies

There are many severe dust explosions in South Korea, and these accidents occur not only in chemical plants but also in general storage facilities. Through three case studies, the problems and characteristics of those dust explosions, which caused many casualties and property loss, can be identified.

### 3.1. HDPE (High Density Polyethylene) Explosions

On 14 March, 2013, an explosion occurred in the HDPE silo at Yeosu Industrial Complex, killing six workers during a manhole installation and injuring eleven workers. In addition to human loss, it caused property damage to nearby factories and destroyed three silos at the factory. The accident investigation revealed that HDPE powder was attached to the inner wall of the silo body, which was filled with air, and the bag filter. During the manhole installation work, some powder dust fell off the lower side and created a deposit due to the vibration caused by tools. The primary explosion happened when the flammable material inside the silo was ignited by the welding sparks during the manhole installation. Heat and shock waves generated in the first silo, as shown in Figure 2, and spread to neighboring silos through the piping connected to the outlet of the HDPE powder bag filter, resulting in the secondary explosion.

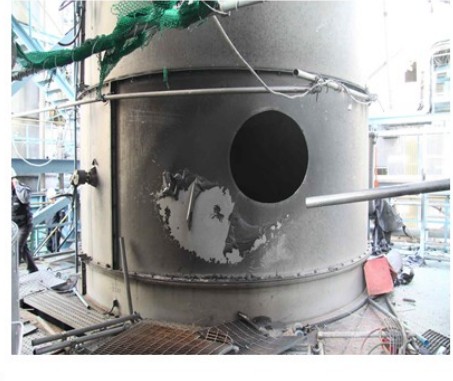 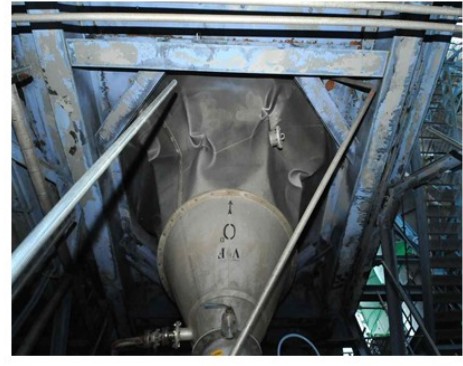

(a)　　　　　　　　　　　　　　　　　　　　　(b)

**Figure 2.** (**a**) The silo where the dust explosion first began; (**b**) Image of silo bottom due to sudden impact.

Details of the accident are as follows: Welding was carried out at six different positions in the silo where the fire was ignited. In general, welding is known to be a very powerful ignition source that ignites all types of explosive mixtures. Aluminum alloy, which the silo was made of, has a thermal conductivity three to five times higher than that of iron alloy. Polyethylene dust attached to the inside of the silo and the bag filter acted as a combustible material and was ignited by welding sparks, resulting in fire. Generally, combustible dust can cause an explosion if the particle size is less than 420 μm. At this time, the polyethylene dust collected inside the silo had an average size of about 60 μm, which is a condition that easily causes an explosion. It is assumed that an explosive environment had been created because of the accumulation of dust and flammable gas. To summarize the progress of the explosion in the silo: (i) The occurrence of fire during welding, (ii) the diffusion on the wall, and (iii) an explosion in the upper position. During the work after the initial gas measurement, continuous ventilation and measurement of gas must be carried out. Unfortunately, the organization's permission to work without ensuring that no explosive mixtures were present in the silo caused the dust explosion.

### 3.2. PP (Polypropylene) Silo Explosions

An explosion occurred inside the polypropylene silo at the Yeosu Industrial Complex, causing a fire in the polypropylene stored inside the silo, resulting in the burning of one aluminum silo on 10 July, 2017. Fortunately, there was no human injury due to the shift time. A fire and explosion, as shown in Figure 3, caused damage to the top of the silo (about 12 m). It is possible that the unreacted components, such as propylene and ethylene impregnated in the polypropylene powders, were desorbed and accumulated in the silo during the synthesis process because the polypropylene was synthesized in a spherical shape around the catalyst. In fact, there was a danger that the flammable gas concentration in the storage silo could increase with time. The flammable gas was measured to 3764 ppm after the fire suppression. Although the gas concentration meter for measuring propylene concentration was present in the process, the calibration of the gas sensor was set without multiplying the gas correction value for methane, and it was mistakenly assumed that it did not exceed the lower explosion limit (LEL) at that time. As the pellet product was conveyed by air pressure, friction between the pipe and the pellet occurred. As a result, friction static electricity was generated and polypropylene dust floating in the silo was charged. Under normal conditions, it was transported to the silo, and even if charged, the residual fine dust was discharged through an air purification device. However, it was presumed that dust with static electricity was introduced into the storage silo, the lower part of which was open at the time of the accident, and the discharged pellet product acted as an ignition source for the flammable gas stagnation space. Estimation of the oxidant suggested that air blowers were used to transport the product to the polypropylene storage silo, and pellets were discharged to the bottom of the accident storage silo, so that air was continuously being introduced through the upper vent pipe.

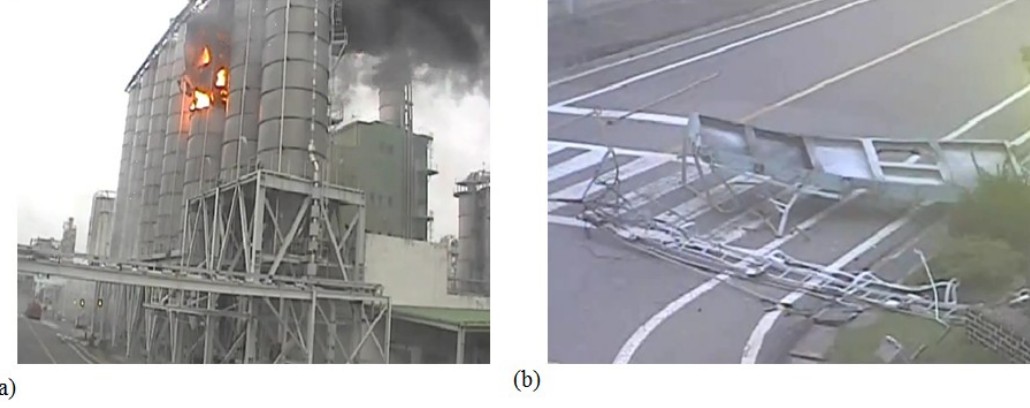

(a)　　　　　　　　　　　　　　　　　　　　　　　　　(b)

**Figure 3.** (**a**) Image of fire after silo explosion; (**b**) Image of upper platform deviating from silo.

### 3.3. Dust Explosion by Ammonium Perchlorate

An accident occurred in a combustion chamber while charging slurry ammonium perchlorate propellant in the mixer ball into the combustion tube, causing a disastrous explosion. This accident occurred on 29 May, 2018, in the Daejeon area, as shown in Figure 4. A sudden fire occurred in the propellant, resulting in the deaths of five workers and injuries of four workers, who were working on site. The explosion occurred during the opening of the discharge valve of the mixer bowl, which was the preparation of the propellant charge.

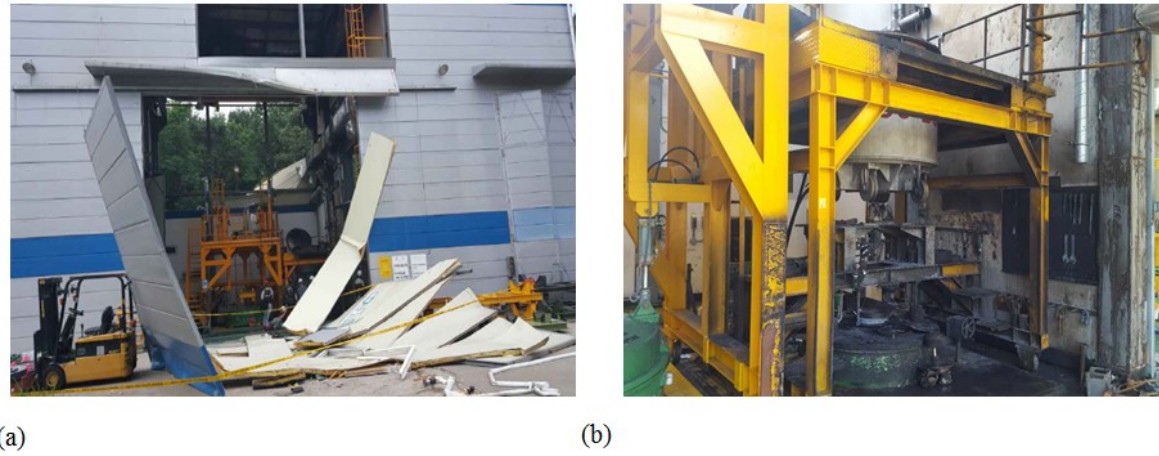

(a)  (b)

**Figure 4.** (**a**) Front image of the place where the accident occurred; (**b**) Image of accident equipment.

The accident investigation was carried out through analysis of the three basic elements of fire. The propellant used a binder that had such physical properties as plastic or rubber, and was intended to bond the components of the propellant to each other. It can also be used as a fuel when burning. The propellant contained 86% ammonium perchlorate, which was used as a powerful oxidizer, and can possibly expand damage in case of a fire. Sources of energy that can act as ignition sources include thermal energy (i.e., fire, high temperature surface), mechanical energy (i.e., friction, shock), and electrical energy (i.e., static electricity, electrical spark). As a result of the accident investigation, it was found that the discharge valve was not open during the charging of the combustion tube, so a worker used a rubber hammer to open the discharge valve. It was estimated that a fire occurred in the propellant due to the impact of hitting the wooden rod on the discharge valve head with the rubber hammer inside the mixer ball containing the propellant. Therefore, it was presumed that an impulse exceeding the propellant impact sensitivity was generated in the slurry propellant by the impact, and a fire occurred in the propellant.

## 4. Discussion

### 4.1. Dust Explosion for Other Countries

Figure 5 shows the results of 87 cases of dust explosions in Japan from 1987 to 2003 [27]. Approximately 51% of dust explosions were caused by metal dust, followed by 17% of the accidents being caused by wood. Dust explosions occurred in 16% and 8% of the accidents caused by plastic and food, respectively. According to the Japan Institute for Occupational Safety and Health (JNIOSH), metal, wood, and plastic dust account for about 85% of the dust explosions.

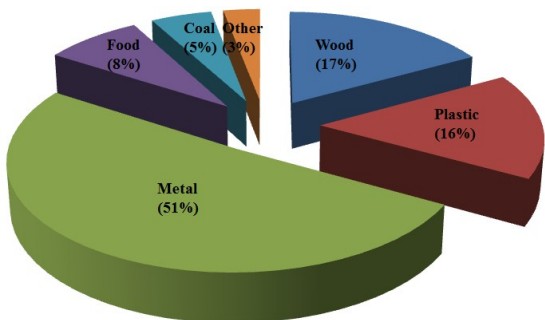

**Figure 5.** Cause of dust explosions in Japan from 1987 to 2003.

Figure 6a presents combustible dust fires and explosions from 1980 to 2005 in the United States. They identified 119 deaths in 78 of the 281 dust explosions [28]. An average of 5 deaths, 29 injuries, and 10 dust explosions occurred annually, according to CSB (Chemical Safety Board) statistical analysis. Although no property damages have been reported by governments or private entities, FM Global Insurance Company reported over 1 million dollars of property losses from 22 dust explosions from 1983 to 2003 [29]. According to the distribution of combustible dust explosions, food products, wood, and metals account for more than 65% of explosions, and plastic accounts for 14%. More than 73% of the accidents were caused by combustible dusts. The CSB also reported that equipment malfunctions caused by dust collectors were the most common accidents in all industries. Figure 6b classifies the dust generated in industries in the United States for two years, with more than 60% in combining food and wood products among 176 cases [30]. In 2016, a total of 31 accidents occurred. With respect to combustible materials, wood and food products accounted for more than 60%, and metal and coal accounted for 9%. Analysis of the dust explosions from equipment showed that 19% occurred in dust collectors, 16% in storage silos/hoppers, and 10% in elevators/conveyors. In 2017, 145 cases were reported, and the number of dust explosions increased, but blowout panels were installed in equipment to make smaller explosions and reduce disastrous explosions. The classification of combustible materials that caused the accidents was similar to that of 2016, with wood and food products accounting for more than 55%, and metal accidents by 8%. However, the type of equipment in the accidents was slightly different from 2016, with 17% in dust collectors, 14% in storage silos/hoppers, and 8% in elevators/conveyors. From Figure 6b, it can be seen that the ratio of dust explosions from food products has increased, but dust explosions from metal have decreased, compared to Figure 6a. Detailed statistics information is provided in Supplementary Information (SI.1 and SI.2).

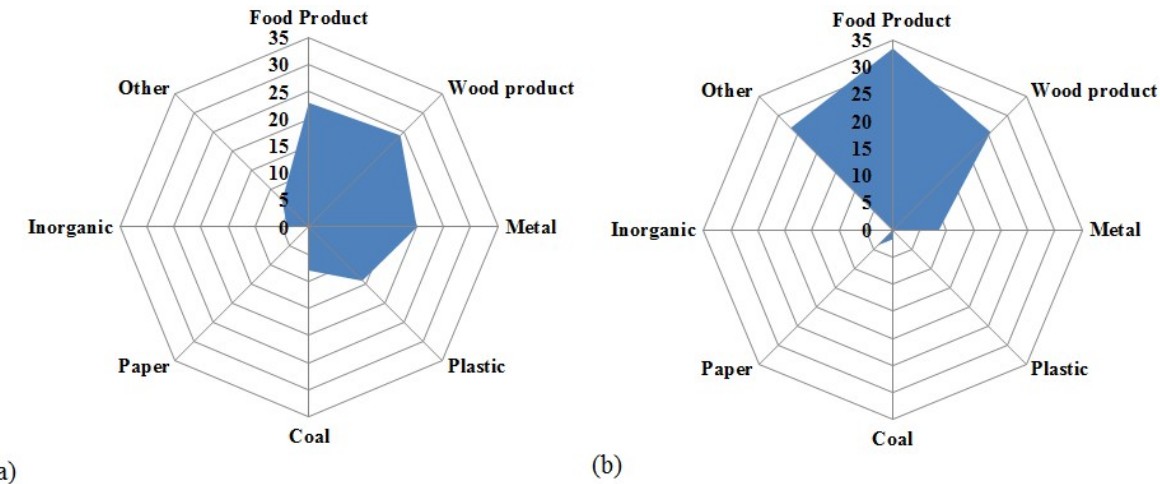

**Figure 6.** (**a**) Distribution of types of dust explosions in the United States (%) from 1980 to 2005; (**b**) Distribution of types of dust occurring in the United States (%) from 2016 to 2017.

Figure 7 showed the analysis of 303 dust explosions in the United Kingdom from 1979 to 1988 [31]. Sixty-nine cases by paper/wood, 55 cases by metal dusts, and 43 cases by food products were reported, as shown in Figure 7a. Those materials caused more than 50% of dust explosions. In addition, 227 cases (74% of 303 cases) were caused by combustible dusts. Analysis of 303 dust explosions with respect to equipment, as shown Figure 7b, revealed that 53 cases of accidents occurred in a mill/grinder, 47 cases of accidents in a filter, and 43 cases of accidents in a dryer. Meanwhile, 19 cases of dust explosions in a silo/hopper accounted for 6.3% of total accidents.

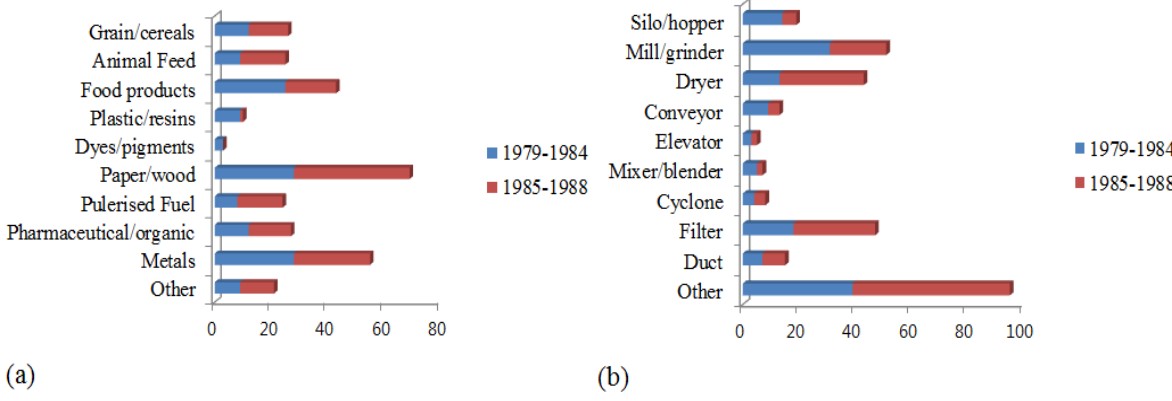

**Figure 7.** (**a**) Distribution of types of dust explosions occurring in the United Kingdom; (**b**) Distribution of dust explosion by equipment.

Figure 8 shows 37 cases of dust explosions from 2000 to 2018 in France. Wood products and plastic dusts accounted for half of the dust explosions, followed by metal and food products at 19% each [32]. More than 50% of dust explosions occurred in silos/hoppers and filters/dust collectors. About 90% or more of accidents were caused by combustible dusts. Compared with the previous Japanese accident analysis, this shows a similar characteristic for dust explosions.

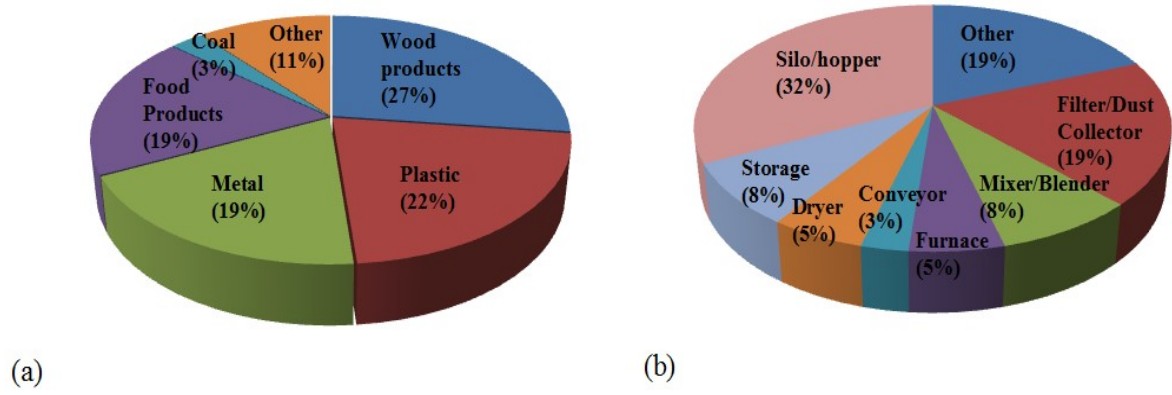

**Figure 8.** Dust explosions in France over 18 years. (**a**) Distribution of types of dust involved in explosion; (**b**) Distribution of dust explosion by equipment.

## 4.2. Characteristics of Dust Explosions between South Korean and Other Countries

Accident analysis of dust explosions in South Korea and other countries found that the causes and types of industry differ from country to country. With the development of the heavy chemical industries, dust explosions were mainly caused by metal, plastic, and food. The main cause of dust explosions in Japan exhibited a similar pattern to South Korea. It is estimated that the geographic proximity and industrial structure are similar to those of South Korea. However, special care must be taken for dust explosions from metals, since the explosiveness of metallic dust can also be an important cause of dust explosions. On the other hand, the sources of dust explosions in the United States and

the United Kingdom were food, wood/paper, and metallic materials, although they occurred with different frequency. The main sources of dust explosions in France were wood, plastic, and metal. As the recent dust explosions in silos were mentioned in the case studies, many dust explosions were caused by equipment, such as silos/hoppers, filters/dust collectors, and mills/grinders in other countries. The workplace should strive to prevent dust explosions and train workers before use to raise safety awareness about the risk of dust explosions.

*4.3. Effective Protective/Mitigatory Measures of Dust Explosions*

Protective/mitigatory measures of dust explosions play a significant role in minimizing human and property losses. As of today, explosion isolation, partial inerting, explosion venting, explosion suppression, and preventing secondary explosions outside process equipment are well-known preventive measures.

Explosion isolation is an important prevention technique to prevent spreading from a primary explosion to other processes. Lunn et al. (1996) conducted a coal dust explosion experiment in a system of linked vessels connected by a pipe [33]. Holbrow et al. (1999) conducted a similar experiment and presented a quantitative guide for design and protective methods, such as explosion containment and explosion venting in interconnected process systems [34]. Taveau (2017) reported that large-scale testing required verifying explosion isolation systems and needed to place the explosion isolation device in the correct position [35]. Partial inerting decreases the oxygen concentration in the atmosphere, rather than the complete removal of oxygen, thereby reducing both the ignition sensitivity and the combustion rate of the dust cloud, as well as substantially increasing the minimum ignition energy (MIE) [36]. Several studies have been conducted experimentally on the effect of oxygen content of MIE on various types of dust. Ackroyd et al. (2011) measured the MIE of organic powders at reduced oxygen concentration and correlated it with the oxygen concentration of the atmosphere [37]. Choi et al. (2015) reported that MIE increased in all the powders used in the experiment as the N2 in the air increased, though it varied depending on the type of powder [38]. Explosion venting is the most efficient and widely used method to mitigate the effects of dust explosions, and many studies have been conducted on it. Tascon et al. (2016) carried out dust explosion experiments. Four vessels with different geometries in a silo were used to analyze the effect of the length/diameter (L/D) ratio, and it was recommended to increase the vent area in elongated vessels when L/D is greater than 1 [39]. Tascon (2017) analyzed the effectiveness of inertia in three different venting systems and presented practical recommendations for vent area design in silos [40]. Explosions in industrial plants usually occur mainly in tanks, reactors, and technical systems, and then extend to the entire system before emerging outside. Active explosion suppression for mitigation of dust explosion is complex and expensive, and is used only when a simple and inexpensive method does not apply. An effective way to minimize the loss of these explosions was to suppress the explosion in the plant and localize the losses. Moore (1987) developed automatic explosion suppression systems and designed early detection of developing explosions. Dust explosions usually last from several tens to several hundred milliseconds in a tank that has a volume of several cubic meters. The extinguishing agent was sprayed at a high speed of about 100 m/s to suppress the explosion for several milliseconds from the beginning to prevent dust explosions [41]. Recently, active explosion suppression using extinguishing powders or water in occupational safety zones is becoming more widespread [42,43]. Based on computational fluid dynamics (CFD), Morgan (2000) confirmed the applicability of numerical models of complex explosion suppression processes and designed novel suppressant injection nozzles [44]. The remaining issues of all efforts to fight the hazards of dust explosions are preventing secondary explosions outside process equipment and are closely related to the impact of the accidents. Cybulski et al. (1993) reported that relatively weak secondary dust explosions in short and narrow tunnels in grain elevators were extinguished by an appropriate design and triggered by water barriers [45]. Eckhoff (2003) claimed that extensive secondary explosions could be removed by freeing dust from outside process equipment [10].

Although many studies are in progress, efficient design of venting arrangements is one of the hardest tasks to prevent propagation of dust explosions in complex coupled-process systems.

## 5. Conclusions

Since dust explosions cause not only human and property loss, but also secondary dust explosions, this study investigated various cases of dust explosions that occurred in South Korea and several other countries. It was found that not only the cause of dust, but also the industrial structure changed as time passed. Therefore, these situations should be identified, and appropriate preventive or mitigating methods should be presented.

In this study, 53 combustible dust- and flammable solid-related fire explosions from 1984 to 2018 in South Korea were reported, with the causes of the dust explosions being similar to those in Japan. For instance, metal dust accounted for 44%, and plastic and food causes accounted for 19%, which was considered to reflect industrial characteristics. There were 176 cases of accidents in the United States between 2016 and 2017, and more than 60% were caused by food and wood products. In 2016, a total of 31 accidents occurred. Analysis of the dust explosion equipment showed that 30% occurred in dust collectors, 12% in storage silos, and 6% in grinding equipment. In 2017, 145 cases were reported, and the number of dust explosions increased, but blowout panels were installed in equipment to make a smaller explosion and reduce disastrous explosion. The equipment that caused the accidents was identified as 18% being dust collectors, 14% storage silos, and 8% elevators/conveyors. There were 303 dust explosions in the United Kingdom from 1979 to 1988. Among them, 53 cases of dust explosions occurred in mills/grinders, 47 cases of accidents in filters, 43 cases in dryers, and 19 cases occurred in silos/hoppers, which are increasingly used in South Korea. Dust explosions from 2000 to 2018 in France showed that wood products and plastic dusts accounted for half of the dust explosions, followed by metal and food products at 19% each. Dust explosions during the last three years were caused by organic matters and metal, and unfortunately, dust explosions have occurred repeatedly during grinding, mixing, and injection of powder materials into facilities. There were no accidents reported during the production process of wood or paper during the last three years.

Dust explosions represent catastrophic accidents, but can be mitigated easily if the mechanism of the dust explosion is identified. Therefore, understanding of basic knowledge contributes to reducing the risk of dust explosions. In addition, prevention measures for dust explosions should be presented, based on recent accident characteristics and the complexity of the process. Taking these facts into account, the following are measures to prevent dust explosions in chemical plants where dust explosions mainly occur:

(i)    Good housekeeping is necessary to prevent accumulation and scattering of dust on the floor of the building;

(ii)   Dust generating facilities should be improved by means of installment of the lid or a sealed structure so that dust is not scattered to the outside;

(iii)  A metal separator must be installed at the crusher inlet to prevent sparks;

(iv)  All dust generation facilities should be connected to the damping system, and if there is a risk of heat accumulation due to internal fixation, a thermometer should be installed;

(v)   If the gas used to calibrate the sensor differs from the actual measurement of the gas concentration, the manufacturer provides the correction value, and the user must apply the correction value to set the sensor sensitivity and alarm;

(vi)  Silos used in the production and storage of various resin products must be evaluated and monitored for the occurrence of gas concentrations at a normal time, due to process characteristics;

(vii) Oxygen concentrations must be lower than the explosion minimum concentration through the inclusion of an inert gas, such as nitrogen. If a change in oxygen concentration is observed, it is considered to be a dangerous situation, and the operation should be immediately stopped until the problem is resolved;

(viii) High-speed operation valves, explosion pressure vents, and explosion suppression devices should be installed to protect against dust explosions.

**Supplementary Materials:** The following are available online at http://www.mdpi.com/2071-1050/11/18/4888/s1, Title: Case Studies for Dangerous Dust Explosions in South Korea during Recent Years.

**Author Contributions:** Conceptualization, C.K., S.J.; methodology, S.P., C.R.; formal analysis, S.J., C.R.; writing—original draft preparation, C.K., S.J.

**Funding:** This research received no external funding.

**Acknowledgments:** This paper cites data from the Korea Occupational Safety and Health Agency (KOSHA) and the Korea Ministry of Employment and Labor.

**Conflicts of Interest:** The authors declare no conflict of interest.

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
