# Peer review of "Case Studies for Dangerous Dust Explosions in South Korea during Recent Years"

_sustainability, doi:10.3390/su11184888_

Round 1
Reviewer 1 Report
1. At the outset I wish to emphasize that any good review of dust explosion accidents in a specific country should be welcomed. Unfortunately the most effective way of fighting the dust explosion hazard still seems to be to learn from the accidents that nevertheless occur over and over again.
2. However, I have found it quite difficult to review this paper. One reason is that the English language needs some brushing-up throughout the paper. This is not only a matter of linguistics, but also a matter of lack of logics.
3. To give this criticism some specific substance, I start with the ABSTRACT. In page line 18 ".. that handle combustible materials .." does not cover the reality and should rather be ".. processes that produce and handle combustible materials in the form of powders and dusts, either as a main product, or as an undesired bi-product," . At the end of the same line "a constant threat" should be replaced by "a constant dust explosion threat".
In page line 20 "over 30 years" to be replaced by "over the last 30 years" or another more precise 30-year time span.
I then move to the INTRODUCTION. In page line 41 ".....where dust spread in the air with finely ground...." does not make much sense to me. The following sentences in lines 42-44 are not very clear either.
The rather simplifying statement in page lines 50 and 51 "According to NFPA 68 aluminium is classified as ST3..." is not valid as a general statement because it does not reflect the strong influence of particle size (see ref. 41 chapter 1, figs. 1.23, 1.28 and 1.30) on both ignitability and explosibility of dust clouds in air at large, including aluminium.
It is not possible for me to go through the entire paper in this way, but I trust that the authors
4. There must be correspondence between the main paper title and the paper content. The title says that the paper covers dangerous dust explosions in South Korea during recent years. As a consequence I suggest that the heading of chapter 2 be "Dust explosions in South Korea 1984-2018." In line with this, it would strengthen the paper if section 2.2 be either deleted or moved to the discussion chapter 4. in the last part of the paper. Section 2.3 would the become 2.2. The condition for keeping the present section 2.2 in the discussion part of the paper in this way is that its content be discussed in relation to the findings for South Korea.
With regard to the references, I suggest that ref. 41 be cited at the beginning of the paper rather than at the end. The reason is that Ref. 41 contains a lot of material that is relevant for the first part of the paper. In ref. 11 the name of the fourth author is missing (page line 383). All references should be checked carefully to make sure that they are correct and complete.
5. This paper contains a lot of valuable information on dust explosions in South Korea during a recent period of 30 years. It is important to get this information published.
Author Response
Response to Reviewer 1
Q1) At the outset I wish to emphasize that any good review of dust explosion accidents in a specific country should be welcomed. Unfortunately the most effective way of fighting the dust explosion hazard still seems to be to learn from the accidents that nevertheless occur over and over again.
Response) We totally agree with your opinion. It is very important to prevent similar dust explosion accidents through lessons from the previous accidents. In South Korea, chemical accidents were analyzed, but dust explosion accidents were not analyzed.
Q2) However, I have found it quite difficult to review this paper. One reason is that the English language needs some brushing-up throughout the paper. This is not only a matter of linguistics, but also a matter of lack of logics.
Response) We received English correction service from Editage Company (Marked as a green color) according to your opinion, and reinforced the logical lacking as much as possible based on the comments you suggested.
Q3) To give this criticism some specific substance, I start with the ABSTRACT. In page line 18 ".. that handle combustible materials .." does not cover the reality and should rather be ".. processes that produce and handle combustible materials in the form of powders and dusts, either as a main product, or as an undesired bi-product," . At the end of the same line "a constant threat" should be replaced by "a constant dust explosion threat".
Response) Thank you for your comment. I followed your recommendation (Line 18~20).
Q4.1) In page line 20 "over 30 years" to be replaced by "over the last 30 years" or another more precise 30-year time span.
Response) Thank you for your comment. I followed your recommendation (Line 21).
I then move to the INTRODUCTION. In page line 41 ".....where dust spread in the air with Q4.2) finely ground...." does not make much sense to me. The following sentences in lines 42-44 are not very clear either.
Response) We modified the sentences you mentioned (Line 40~43).
Q4.3) The rather simplifying statement in page lines 50 and 51 "According to NFPA 68 aluminium is classified as ST3..." is not valid as a general statement because it does not reflect the strong influence of particle size (see ref. 41 chapter 1, figs. 1.23, 1.28 and 1.30) on both ignitability and explosibility of dust clouds in air at large, including aluminium.
Response) Thank you for your comment. We deleted the sentences as you mentioned and described the effect of particle size on the flammability and explosiveness of dust clouds in the air (Line 52~57).
Q4.4) It is not possible for me to go through the entire paper in this way, but I trust that the authors
Response) First of all, we are very sorry for you to have trouble understanding this manuscript. We endeavored to make the manuscript an improved version by means of proofreading and solving logical problems.
Q5.1) There must be correspondence between the main paper title and the paper content. The title says that the paper covers dangerous dust explosions in South Korea during recent years. As a consequence I suggest that the heading of chapter 2 be "Dust explosions in South Korea 1984-2018." In line with this, it would strengthen the paper if section 2.2 be either deleted or moved to the discussion chapter 4. in the last part of the paper. Section 2.3 would the become 2.2. The condition for keeping the present section 2.2 in the discussion part of the paper in this way is that its content be discussed in relation to the findings for South Korea.
Response) Thank for your suggestion. We followed your recommendation and corrected the subtitles. Section 2.2 moved to the discussion section in Section 4.1. Section 2.3 has been changed to Section 2.2. We added Section 4.2 to further describe characteristics of dust explosions between South Korean and other countries.
With regard to the references, I suggest that ref. 41 be cited at the beginning of the paper rather than at the end. i) The reason is that Ref. 41 contains a lot of material that is relevant for the first part of the paper. ii) In ref. 11 the name of the fourth author is missing (page line 383). iii) All references should be checked carefully to make sure that they are correct and complete.
Response) We followed your recommendation; i) Ref. 41 moved to Ref. 10. ii) we modified our mistake, iii) all references were check again
Q6) This paper contains a lot of valuable information on dust explosions in South Korea during a recent period of 30 years. It is important to get this information published.1.
Response) Thank you for your review comments. We sincerely hope that the logical problems in this manuscript are well resolved.
Reviewer 2 Report
This paper reports about the 53 dust explosions that occurred in South Korea over the last 30 years. It analyses the causes in general with emphasis on the recent years and compares them to dust explosions that occurred in various other countries. The content is very important as it is highly appreciated to learn from such accidents with the aim to avoid them in future. This will save either lives, or money and environmental damages. Thus, the paper is undoubtful worth to be published. The paper is well written and presents the results in a comprehensive manner. Well done!
I encourage the authors to extend their statistical analysis and conclusions even more to the technical and organisational origins and backgrounds of the accidents. It will be much more effective to know the detailed causes, e.g. the specific malfunctions of the equipment, the particular ignition sources in conjunction to the hazards and the operational mismanagement as well as the operational mistakes or lack of information of the staff in charge. This could also deliver an input to national and/or international regulations and standards.
Some editorial remarks:
Line 49: The st of Kst should be an index Line 89: [22] should be [20,22] Line 112: According to dust human loss caused by dust explosions… should be According to human loss caused by dust explosions… (delete the first dust) Line 132: ...reported over 1 million of property losses... (a reference is missing) Figure 3 and lines 147-148: This statement has no statistical evidence since the data are not from the same statistical population. Table 1: The headline “Ignition Source” is misleading and should be e.g. “Hazardous Material”. Page 6, last para (lines 195-208): Welding is a very strong ignition source that will ignite all types of explosive mixtures. Thus, it does not depend on the hazardous material. An organizational permission of the work would have been necessary after ensuring that no explosive mixture is present.Author Response
Response to Reviewer 2
Q1) This paper reports about the 53 dust explosions that occurred in South Korea over the last 30 years. It analyses the causes in general with emphasis on the recent years and compares them to dust explosions that occurred in various other countries. The content is very important as it is highly appreciated to learn from such accidents with the aim to avoid them in future. This will save either lives, or money and environmental damages. Thus, the paper is undoubtful worth to be published. The paper is well written and presents the results in a comprehensive manner. Well done!
Response) Thank you for your comment.
Q2) I encourage the authors to extend their statistical analysis and conclusions even more to the technical and organisational origins and backgrounds of the accidents. It will be much more effective to know the detailed causes, e.g. the specific malfunctions of the equipment, the particular ignition sources in conjunction to the hazards and the operational mismanagement as well as the operational mistakes or lack of information of the staff in charge. This could also deliver an input to national and/or international regulations and standards.
Response) We totally agree with your comment. In recent dust explosions, the cause of the dust explosions can be well analyzed. Please note that it is almost impossible to obtain detail information on the dust explosion before 2005. Some information is missing. We wanted to provide the specific malfunctions of the equipment as well as statistical analysis.
Some editorial remarks:
Q3) Line 49: The st of Kst should be an index
Response) We modified our mistake (Line 51).
Q4) Line 89: [22] should be [20,22]
Response) We modified our mistake (Line 90).
Q5) Line 112: According to dust human loss caused by dust explosions… should be According to human loss caused by dust explosions… (delete the first dust)
Response) We modified our mistake (Line 113~114).
Q6) Line 132: ...reported over 1 million of property losses... (a reference is missing)
Response) We added the reference you mentioned (Line 223).
Q7) Figure 3 and lines 147-148: This statement has no statistical evidence since the data are not from the same statistical population.
Response) We fully agree with your opinion. We modified the sentence you mentioned and provided statistical information in the Supplementary Information 1 (SI. 1~SI.2)
Q8) Table 1: The headline “Ignition Source” is misleading and should be e.g. “Hazardous Material”.
Response) We followed your recommendation (Table: Line 127).
Q9) Page 6, last para (lines 195-208): Welding is a very strong ignition source that will ignite all types of explosive mixtures. Thus, it does not depend on the hazardous material. An organizational permission of the work would have been necessary after ensuring that no explosive mixture is present.
Response) We agree with your opinion. We included your opinion in this manuscript (Line 144~146, Line 155~157).
Round 2
Reviewer 1 Report
I have now reviewed the revised version of the paper "Case studies for dangerous dust explosions in South Korea during recent years" and recommend that the revised version be published in your journal.
Author Response
Q1) I have now reviewed the revised version of the paper "Case studies for dangerous dust explosions in South Korea during recent years" and recommend that the revised version be published in your journal.
Response) Thank you again for your review comments throughout the revision process that helped us to improve the quality of the manuscript.